# Machine Learning-Enabled Quantitative Analysis of Optically Obscure Scratches on Nickel-Plated Additively Manufactured (AM) Samples

**DOI:** 10.3390/ma16186301

**Published:** 2023-09-20

**Authors:** Betelhiem N. Mengesha, Andrew C. Grizzle, Wondwosen Demisse, Kate L. Klein, Amy Elliott, Pawan Tyagi

**Affiliations:** 1Mechanical Engineering, University of the District of Columbia, Washington, DC 20008, USAandrew.grizzle@udc.edu (A.C.G.); wondwosen.demisse@udc.edu (W.D.); kate.klein@udc.edu (K.L.K.); 2Manufacturing Demonstration Facility, 2350 Cherahala Boulevard, Knoxville, TN 37932, USA; elliottam@ornl.gov

**Keywords:** unsupervised machine learning, K-means clustering, additive manufacturing, nickel plating, hardness, scratch test

## Abstract

Additively manufactured metal components often have rough and uneven surfaces, necessitating post-processing and surface polishing. Hardness is a critical characteristic that affects overall component properties, including wear. This study employed K-means unsupervised machine learning to explore the relationship between the relative surface hardness and scratch width of electroless nickel plating on additively manufactured composite components. The Taguchi design of experiment (TDOE) L9 orthogonal array facilitated experimentation with various factors and levels. Initially, a digital light microscope was used for 3D surface mapping and scratch width quantification. However, the microscope struggled with the reflections from the shiny Ni-plating and scatter from small scratches. To overcome this, a scanning electron microscope (SEM) generated grayscale images and 3D height maps of the scratched Ni-plating, thus enabling the precise characterization of scratch widths. Optical identification of the scratch regions and quantification were accomplished using Python code with a K-means machine-learning clustering algorithm. The TDOE yielded distinct Ni-plating hardness levels for the nine samples, while an increased scratch force showed a non-linear impact on scratch widths. The enhanced surface quality resulting from Ni coatings will have significant implications in various industrial applications, and it will play a pivotal role in future metal and alloy surface engineering.

## 1. Introduction

Machine learning (ML) has received a great deal of attention recently, particularly as a result of recent developments in the field of deep learning [1,2]. Artificial intelligence has become a central focus across various research fields and in additive manufacturing, like engineering disciplines [3,4,5,6], as it offers a unified framework through which to integrate intelligent decision making into numerous fields [7,8]. There are several forms of additive manufacturing, such as binder-jetting-based metal additive manufacturing (BJAM). In this study, we focused on stainless-steel and bronze composite samples that were manufactured using the binder-jetted method [9]. Binder jetting is a 3D printing process that involves the deposition of an adhesive binding agent onto thin layers of powdered material. The printer head moves over the build platform depositing binder droplets, and it then prints each layer in a way that is not dissimilar to 2D printers that print ink on paper. After each layer is complete, the powder bed moves downward, and the printer spreads a new layer of powder onto the build area. The process goes on layer by layer until all parts are complete. After printing, the parts are in a green, or unfinished, state, and they require additional post-processing before they are ready to use. Often, the operator adds an infiltrating substance to improve the mechanical properties of the parts. The infiltrate substance is usually bronze in the case of metal 3D prints [9,10]. The BJAM process has a significant scope of improvement with the application of ML and deep learning [11,12]. The further advancement of AM is critically dependent on the post-processing of completed parts and the use of ML in solving problems.

While BJAM and other additive manufacturing processes have been widely used for rapid prototyping [13,14], some of its constraints revolve around reliability and control [15,16]. Compared to subtractive manufacturing procedures, AM creates objects with poor surface smoothness [9,17]. As produced, surface quality has a negative effect on the tribological behavior of printed parts. It is well understood that rough surfaces tend to experience faster wear compared to smooth surfaces [18,19]. Therefore, it becomes crucial to thoroughly investigate and regulate the surface roughness of AM parts via different approaches to deal with interior and exterior surface quality [20]. The study and control of surface roughness in AM components are essential in order to enhance their durability, reduce wear, and improve overall performance. By understanding and managing the surface roughness, it is possible to optimize the tribological characteristics and extend the lifespan of AM parts [21]. In the context of BJAM, coatings are indispensable for maintaining integrity in a reactive environment. Due to the use of soft and hard materials, a binder-jetted part may be susceptible to corrosion and may display non-uniform mechanical properties. Hence, a protective coating, depending upon the end use, may be a necessity when utilizing BJAM parts.

Among various coatings, electroless nickel coating has been widely applied and studied for conventional engineering components. Electroless nickel plating has been extensively studied with various plating baths so as to identify the optimal conditions for achieving desired qualities such as corrosion resistance, wear resistance, and hardness. Through systematic experimentation and analysis, researchers have aimed to identify the ideal parameters and bath compositions that can lead to electroless nickel coatings with excellent performance in terms of corrosion resistance, wear resistance, and hardness. These efforts contribute to the development and application of electroless nickel plating as a reliable surface treatment method for enhancing the functional properties of various materials [22,23].

In this research, we explored electroless nickel coatings for BJAM parts. The major challenge was experienced in analyzing the hardness of nickel coating via the standard scratch test process. A standard method for determining surface hardness using scratch testing involves running a diamond stylus across the coated surface while applying increasing force until adhesion failure is observed [24]. In the study, it was observed that the surface of the Ni-plated samples exhibited minor scratches, which posed challenges when aiming to accurately capture 3D images with a light microscope. The highly reflective surface resulted in an oversaturation and lens reflection artifacts in the images, making it difficult to quantify scratch widths effectively. To overcome this limitation, scanning electron microscopy (SEM) was employed to generate a 3D height map of the area, thus providing a higher resolution for measuring scratch widths. To address this challenge in the postprocessing of BJAM parts, we applied ML. ML algorithms have proven valuable in addressing various problem-solving tasks such as regression, classification, and forecasting [25]. ML can be broadly categorized into four types based on the learning approach used: supervised, unsupervised, semi-supervised, and reinforcement learning. In unsupervised ML, the algorithm predicts outputs without any explicit supervision, and it relies on unlabeled datasets. One prominent approach in unsupervised ML is clustering, which involves extracting natural groups from data based on their similarities [26,27]. The K-means algorithm is the most well-known and often-used unsupervised clustering method [28]. The K-means cluster seeks to determine the centroid of each cluster and assign the data points to the nearest centroid. The centroid is the arithmetic mean of all the points belonging to the cluster [29]. It iteratively calculates the cluster centroids repeatedly, and adjusts the parameters until a negligible change is observed [30]. There is no need for a training dataset since it is a type of unsupervised ML, and computation is conducted on the real dataset [31].

To the best of our knowledge, for the first time, we explored the application of the K-means ML approach to successfully analyze the scratch on electroless nickel films coated on BJAM. To automate the analysis process and extract scratch data from the images and height maps, a Python script was originally developed. The script utilized the K-means algorithm, an unsupervised machine learning method, to segment and identify the scratches. Applying the K-means algorithm meant that the scratch data could be effectively extracted, thus allowing for a quantitative analysis and characterization of the scratches. The outcomes of this study demonstrated the utility of unsupervised machine learning techniques, such as the K-means algorithm, in addressing challenges encountered in materials science. By leveraging these methods, researchers can overcome limitations in traditional image analysis approaches and obtain valuable insights from complex surface data, such as scratch measurements.

## 2. Materials and Methods

The focus of this paper is on the application of ML-based image analysis approaches for successfully studying scratches that are created on nickel-coated BJAM samples. The BJAM samples used in this study were manufactured by the ExOne^®^ (Huntington, PA, USA). The stainless-steel 420 powder was shaped with a binder jet 3D printer that was made by ExOne^®^. Binder jetting works by spreading powder into a layer, and an inkjet printhead is used to selectively deposit a binder into the layer of powder. As the process proceeds, the powder and binder are layered to form a 3D shape in the powder bed. The average particle size is between 15 and 30 microns, and the binding agent is a polymer. The print is then heated to 200 °C to evaporate the solvent from the binder. Once dried, the parts are removed from the powder bed and set up for post-processing. The post-process consists of adding the part to a crucible filled with a measured amount of bronze alloy. The crucible with the part and bronze is heated to around 1100 °C for 1–2 h, which allows the bronze to melt and infiltrate the porous stainless-steel print. Infiltration is driven by the wetting of the molten bronze and the steel, the surface tension of the molten bronze, and the resulting capillary forces between the stainless-steel particles. Once solidified, the resulting part is a stainless-steel and bronze metal-matrix composite, which is approximately 60% stainless steel and 40% bronze by volume. In the follow-up SEM and EDS analyses with Phenom XL SEM purchased from Nanoscience^®^, (Phoenix, AZ, USA), we observed elemental analysis results that were specific to how many 420 stainless-steel powder particles were present in the imaging area, as well as the variation in the shape and size of each particle. Hence, due to the limitation of EDS in providing consistent results, we report on the percentage of stainless steel and bronze based on the manufacturing process.

Importantly, this paper is mainly about post-manufacturing surface property improvement where the surface properties of BJAM is a critical factor. As a part of post-processing, we developed an empirical model targeting the smooth surface morphology of several micron-thick nickel depositions on nine binder-jetted 420 stainless steel/bronze components. The electroless plating solution was acquired from the Surface Technology Incorporated^®^ company. The experiment plan for nine samples was based on the Taguchi design of experiment, which enables the study of multiple variables and their levels in fewer experiments when compared to the experimental plan where one variable is varied at a time [32]. In this investigation, there were four factors with three levels each. The plating bath solution’s phosphorus levels consisted of low (1–4%), medium (6–9%), and high (10–13%). The temperature levels included low (recommended −10 °C), medium (recommended), and high (recommended +10 °C). For low and medium phosphorus, the recommended temperature was set at 90 ℃, while for high phosphorus, it was 85 ℃. The surface cleaning preparation factor encompassed three levels: organic solution cleaning, plasma cleaning, and chempolishing. Chempolishing-based surface finishing details are published elsewhere [17]. The plasma was produced by 100 W of RF power, at a 30 SCCM Ar flow rate, and at 320 mTorr pressure to etch the binder-jetted samples isotropically. Plasma cleaning was done with SPI Plasma Prep II (West Chester, PA, USA). The fourth factor, plating thickness, also comprised three levels. We targeted depositing at 20, 30, and 40 µm thicknesses, which were determined from the manufacturer-provided data sheet for the three plating solutions. Table 1 depicts the L9 orthogonal array and each sample’s name ID, which are utilized in the discussion section when referring to each sample.

After completing the nickel-plating process as per the plan mentioned in Table 1, scratch testing was performed with a Taber Scratch tester^®^. Scratch geometry analysis is a critical step in determining the toughness of films, and the surface hardness of the composite samples was evaluated by varying the scratch load gradually from 8 N to 15 N. The samples were divided into three groups based on their surface preparation. In general, the trend observed in the graphs indicates that, as the scratch load remains constant, the hardness tends to decrease as the scratch width becomes deeper and wider. This implies that a deeper and wider scratch shows a low surface hardness. Moreover, the relationship between scratch width and applied scratch load is directly proportional, meaning that, as the applied load increases, the scratch width also increases. However, it is important to note that the increase in scratch width is non-linear, suggesting that the relationship between load and width may not be strictly linear. However, since nickel plating makes the surface quite shiny, it became difficult to determine the scratch depth and width profile accurately from an analysis of the optical images. We developed a solution to this problem by relying on the SEM images of the scratches, which produced better depth contrast. The SEM images were visually marked for the location of the scratch, and the K-means machine learning algorithm was applied. The following section describes the K-means algorithm adopted in this study.

K-means clustering is an iterative method that aims to divide a dataset into a predetermined number, K, of distinct clusters or subgroups based on their attributes. The goal is to create clusters that are as dissimilar from each other as possible while making the data points within each cluster as similar as possible. The process begins by randomly assigning K centroids, which serve as the initial center points for the clusters, as shown in Figure 1b. Each data point in the dataset is then assigned to the cluster with the nearest centroid based on a chosen distance metric, typically the Euclidean distance, as shown in Figure 1c. This assignment step ensures that data points are allocated to the group that is closest to them in terms of attribute similarity. After assigning all the data points to clusters, the algorithm recalculates the centroids of each cluster by computing the mean (arithmetic average) of all the data points within the cluster. This updating step adjusts the centroids’ positions to reflect the clusters’ new center points based on the reassigned data points, as shown in Figure 1d. The algorithm iterates between the assignment and update steps until convergence is reached. Convergence is determined by assessing whether there has been a substantial change in the centroids compared to the previous iteration. If the centroids remain largely unchanged, or if the maximum number of iterations is reached, the algorithm terminates. To determine whether a data point belongs to a particular cluster, the algorithm compares the distance between the data point and the centroid of that cluster. Centroid of the cluster is shown by the red and black “x” for two groups in Figure 1b–d. If the distance is less than a certain threshold, which is often represented by the within-cluster sum of squares or a cost function, the data point is assigned to that cluster. Throughout the iterations, the algorithm strives to minimize the cost function by adjusting the positions of the centroids. This process leads to the formation of well-defined clusters that are distinct from each other, with reduced variability within each cluster. The data points within each cluster become more homogeneous or similar to each other in terms of their attributes.

In K-means clustering, the hyper-parameter K is predetermined before the training process begins. The letter “K” represents the number of clusters that the algorithm aims to create. This value is typically determined based on prior knowledge or domain expertise. The objective function in K-means clustering involves minimizing the total within-cluster sum of squares, also known as inertia or distortion. The objective function can be mathematically expressed as follows:(1)$$J=\sum_{j=1}^{k}\sum_{i=1}^{n}{\left\|X_{i}^{j}-C_{j}\right\|}^{2}$$
where *J* is the objective function, and *k* and *n* are the numbers of clusters and cases, respectively. *X* is the case *i*, and *C* is the centroid for cluster *j*. The term in absolute value is known as the distance function.

The number of clusters in the K-means method represent the moving centroids within the data. The elbow method helps determine the optimal number of clusters by evaluating the distortion or inertia for the different values of “K”. The elbow point shown in Figure 2, where the distortion begins to reduce linearly, is chosen as the ideal number of clusters. This method ensures a balance between capturing the right data structure and avoiding overfitting.

## 3. Results and Discussion

The surface hardness of the composite samples was evaluated by conducting a scratch test, where the scratch load was gradually increased from 8 N to 15 N. The continuous and highly reflective films did not exhibit noticeable micropores. The samples were divided into three groups based on their surface preparation. In general, the trend observed in the graphs indicated that, as the scratch load remains constant, the hardness tends to decrease as the scratch width becomes wider. This implies that a wider scratch shows a low surface hardness. Moreover, the relationship between scratch width and applied scratch load is directly proportional, meaning that, as the applied load increases, the scratch width also increases. However, it is important to note that the increase in scratch width is non-linear, suggesting that the relationship between load and width may not be strictly linear.

The process of quantifying the scratch width is depicted in Figure 3. In Figure 3b, the shaded area represents the region identified as the scratch. To accomplish this, an individual performed the shading manually using basic image editing software, such as Microsoft Paint. Since the image is in grayscale, consisting of shades of black and white, a K-means clustering algorithm was employed to separate the darker scratched area from the rest of the image. The K-means clustering algorithm is a technique used to partition data into distinct clusters based on their similarity. In this case, it was applied to the grayscale image to create two clusters: one representing black and one representing white. By analyzing the intensity values of the pixels in the image, the algorithm assigned each pixel to one of the two clusters based on its similarity to either black or white.

After the K-means clustering was performed, the resulting clusters provided the coordinates of the pixels within the image that belonged to the black cluster, which represented the scratched area. These coordinates were then utilized to identify the corresponding region on the SEM height map, which provides three-dimensional information about the sample’s surface. Figure 3c illustrates the scratch area on the SEM height map. By using the coordinates obtained from the K-means clustering, the scratched region was precisely located and delineated by a boundary line. This enabled a visual representation of the boundaries of the scratch. Finally, in Figure 3d, a more detailed view of the scratch limits is depicted on the contour plot produced by the Phenom XL SEM 3D reconstruction software. The boundary line clearly indicates the extent and shape of the scratch, thereby providing a comprehensive understanding of the scratch width and its specific location on the sample’s surface.

Once the scratch width data were obtained, they were grouped according to the surface cleaning preparations, as shown in Figure 4. The data were divided into three groups: the first group consisted of samples that underwent chempolishing surface-cleaning preparation, the second group comprised samples that were prepared with organic cleaning, and the third group included samples that were prepared with plasma cleaning. In Figure 4a, the first group is depicted, which contains three samples that underwent chempolishing surface preparation. The CP1 sample shows a scratch width in the ~80–~100 µm range as the load increased from 8 to 12 N. Around 13 N, the scratch width varied, indicating the appearance of more burrs along the scratch contour, thus causing significant jaggedness. Further increase in the load brought the scratch width into the short range (Figure 4a). ML scratch analysis was effective in observing an increase in the average scratch width for the CP2 samples that were subjected to an increasing load (Figure 4a). The average scratch width increased marginally from 60 to 80 µm as load increased 8 to 11N; after that, the load scratch width fluctuated between ~60 to ~100 µm with a large standard deviation that showed the change in material response from a smooth plastic transformation to more burs along the scratch profile. Interestingly, for the 15 N load, a non-uniformity in scratch width was observed, similar to CP1 (Figure 4a). A similar trend was also observed for CP3 as the scratch load increased from 8–15 N. However, for the CP3 sample, the starting average scratch width was around 120 µm for 8 N. This study suggests that the nickel plating hardness on the CP2 sample was around two times more than the plating hardness of the CP3 sample (Figure 1a). By comparing CP1, CP2, and CP3 data, it becomes clear that a significant and clear transition in failure mode occurs between the 12–14 N load range.

The effect of different plating parameters was also studied on the organically cleaned sample (OC group) in Figure 4b. The OC1 sample showed a rather quick jump in average scratch width from the average ~85 µm to the ~120 µm range; the OC1 scratch width remained rather consistent for most of the load range. For the 15 N load, the scratch width was quite non-uniform and appeared with a large variation (Figure 4b). Similarly, the OC2 sample followed the trend observed with OC1. However, the starting scratch width was significantly lower than that observed on OC1. Interestingly, for OC3, the scratch width increased gradually up to 11 N from the ~90 µm to ~120 µm range; after that, the scratch width kept increasing. It appears that for the OC samples, the scratching mechanism was altered in the early stage when compared to the CP samples.

In the case of plasma-treated samples (PC1–3), scratch widths were analyzed. The PC1 sample showed an average scratch width in the ~70 to ~100 µm range as the load increased from 8 to 14 N (Figure 4c). Interestingly, the scratch widths for the PC2 samples increased linearly as the load increased from 8 to 15 N, and the smallest variation was observed in this sample. For the PC3 sample, the scratch width roughly increased with the load. This large variation was attributed to the chempolishing impact on surface morphology because chempolishing can selectively etch one of the components of the BJAM part, thus resulting in a rougher surface. The CP2 samples exhibited a lower average scratch width when compared to CP1 and CP3. This meant that the CP2 samples that had a medium phosphorous (P) nickel coating applied were harder. The CP3 samples showed a significantly larger scratch width with high scattering. It is possible that the nickel coating quality varied significantly when a high-P nickel coating was attempted. Moving on to Figure 4b, the second group represents the three samples that underwent organic cleaning preparation. It is noteworthy that, unlike chempolishing, the organic cleaning process did not impact the BJAM sample surface. Due to better surface smoothness, there was, in general, less scattering. Mid-P nickel coating produced a ~70 µm scratch width, which was nearly 30% lower than the low- and high-phosphorous nickel coatings (Figure 4b). On average, this group was the second hardest, with OC2 (the second organically cleaned sample) showing a high surface hardness that was quite close to the PC2 (the second plasma cleaning) sample. Figure 4c displays the third group, which comprises the samples that underwent plasma cleaning preparation. Plasma cleaning isotopically cleaned the BJAM sample to render a smoother surface. As a result, in general, there was less scattering in the scratch width data. The PC2 samples showed a ~60 µm scratch width, which was clearly more severe than the PC1 samples where low-P solutions were used for Ni coating. Notably, PC2—which represents a combination of a mid-phosphorus level, a temperature 10 degrees lower than the recommended value, and optimal time parameters—demonstrated the highest hardness among all of the samples and had the smallest scratch width. Based on these ML-enabled findings, it is recommended to utilize plasma and organic cleaning methods when aiming for a harder surface. The plasma cleaning method, particularly represented by PC2, resulted in the hardest surface, while the organic cleaning method, particularly represented by OC2, showed a relatively high surface hardness comparable to PC2. Therefore, for applications where a harder surface is desired, the utilization of plasma and organic cleaning methods is recommended based on an analysis of the scratch width and surface hardness data.

## 4. Conclusions

The K-means unsupervised ML algorithm was employed to address the challenges associated with optically obscure scratches on nickel-plated AM samples. In this context, the samples were prepared using the L9 orthogonal array TDOE methodology. Due to the nature of Ni-electroless plating, some of the samples exhibited a shiny appearance, making them difficult to analyze accurately when using a digital light microscope. This was primarily due to issues such as light saturation and reflection. To overcome this, SEM was utilized to generate grayscale images and corresponding 3D height maps of the scratched Ni-plating surfaces. Subsequently, the K-means ML clustering approach was applied to visually detect the scratch areas within the SEM images. Through this approach, it was observed that the TDOE methodology resulted in distinct levels of Ni-plating hardness for each of the nine samples. Furthermore, as the scratch force increased, the scratch widths exhibited a non-linear increase, thus highlighting the complex relationship between applied force and scratch width. Our image analysis capabilities highlighted that mid-P nickel coating produced harder coating when compared to low- and high-P content-based nickel coatings. This study also showed that the chempolishing treatment on BJAM produces a higher roughness that impacts the uniformity and quality of nickel coatings. Our research suggests that surface preparation must be chosen with great care to target the specific attributes of electroless nickel coatings, and microscopic high-resolution SEM images should be considered for an adequate understanding of the morphologies that evolve due to interacting parameters. A scratch width analysis with a Taguchi design of experiment should be focused on specific properties. The CP2, PC2, and OC2 samples, where the medium-phosphorous solution was used, appeared to yield harder coatings. Our ML-enabled scratch width analysis was able to capture the differences in various factors leading to the differences in scratch widths and deviations. The difference in standard deviations at each load for each sample category was reflective of the difference in the surface microstructure after different processing techniques and electroless nickel coatings were applied. The K-means clustering approach utilized in this work was able to capture the variation in load. The demonstrated methodology of combining SEM imaging, K-means clustering, and scratch width quantification offered a practical solution in surface analysis when faced with obstacles such as optically obscure scratches. In future work, different clustering and ML approaches may be applied to analyze scratch widths.

## Figures and Tables

**Figure 1 materials-16-06301-f001:**
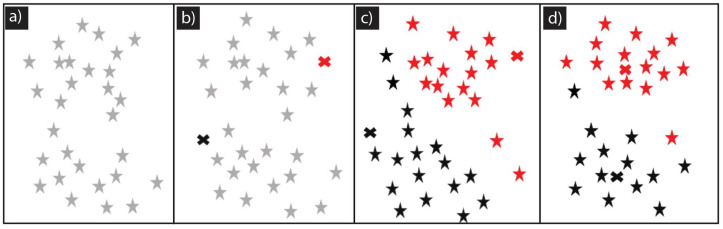
K-means clustering process illustration: (**a**) row data, (**b**) random initializing centroids to represent the center of a cluster, (**c**) for each data point, calculate its distance to each centroid and assign data points to the nearest centroid, and (**d**) update the centroids by computing the mean of all data points assigned to each cluster and repeat until convergence.

**Figure 2 materials-16-06301-f002:**
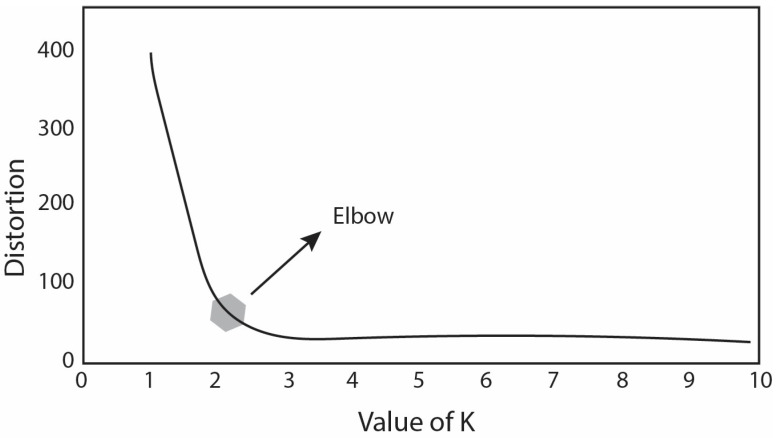
The elbow method being used to find the best K value for the K-means clustering in this paper.

**Figure 3 materials-16-06301-f003:**
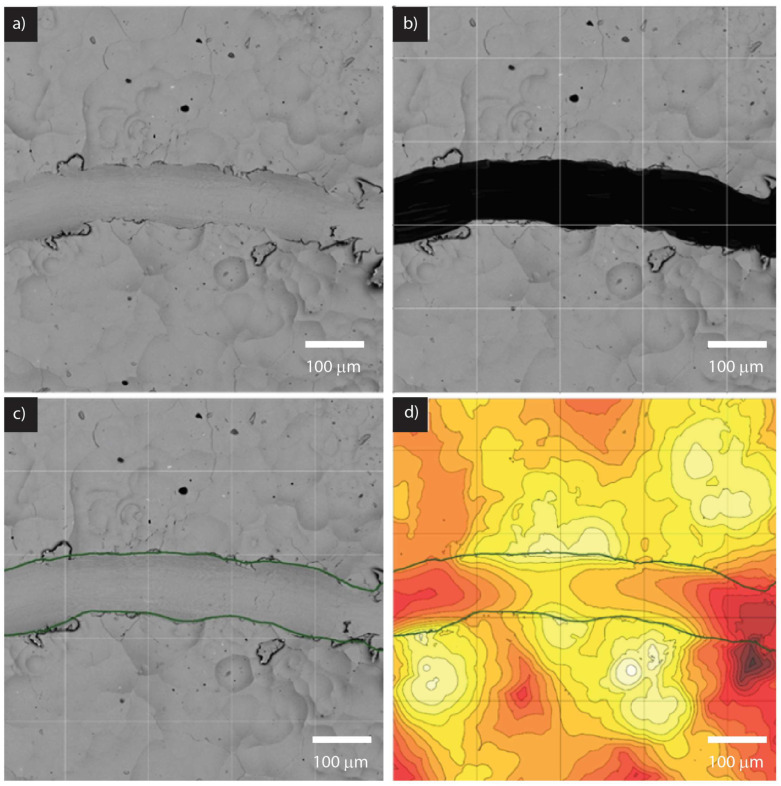
SEM images of the Ni-plated composite surface. (**a**) Raw image obtained using a full backscatter detector (BSD), (**b**) shaded section used to differentiate between scratches and Ni-plating, (**c**) scratch mask coordinates obtained using K-means clustering of the shaded section in (**b**), and (**d**) a contour map of the height map obtained from SEM with a scratch mask for quantifying scratch width.

**Figure 4 materials-16-06301-f004:**
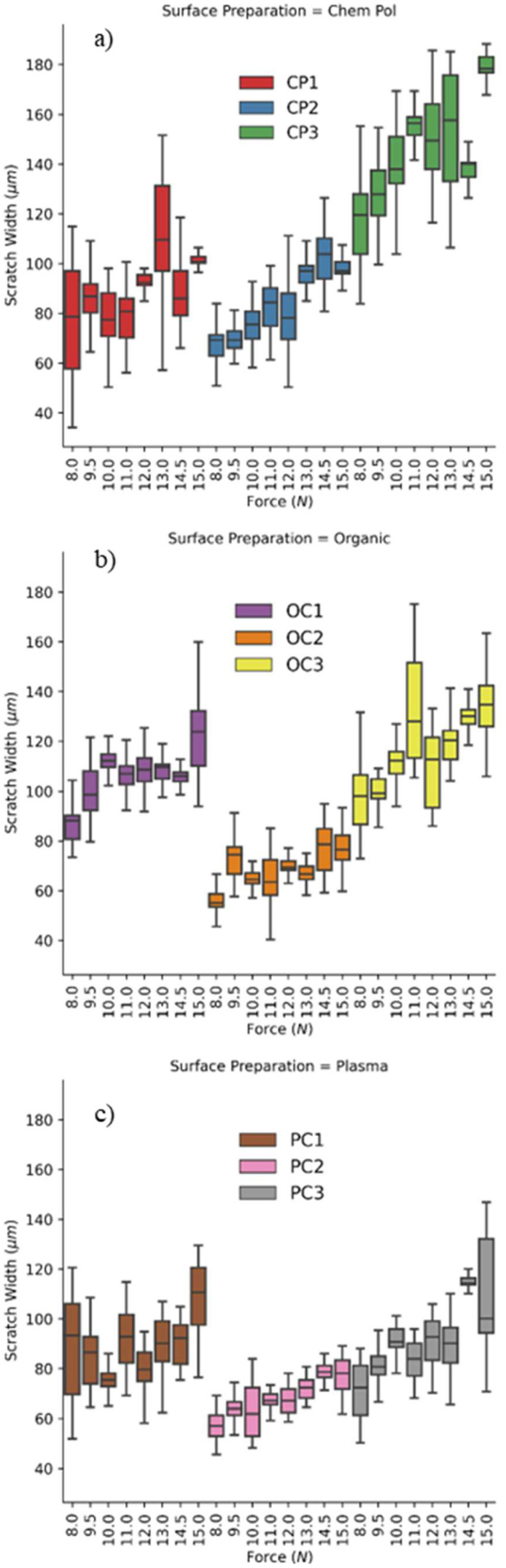
Scratch width measured after a K-means clustering of the samples with different scratch loads: (**a**) samples prepared by chempolishing, (**b**) samples prepared with organic cleaning, and (**c**) samples prepared with plasma cleaning.

**Table 1 materials-16-06301-t001:** The L9 orthogonal array of the nine experiments for investigating nickel plating.

Exp. Run	Phosph. Level	Temp. (°C)	Surface Prep.	Thickness (µm)	ID
1	Low	85	Organic	20	OC1
2	Low	95	Plasma	30	PC1
3	Low	105	Chempolish	40	CP1
4	Medium	85	Plasma	40	PC2
5	Medium	95	Chempolish	20	CP2
6	Medium	105	Organic	30	OC2
7	High	75	Chempolish	20	CP3
8	High	85	Organic	30	OC3
9	High	95	Plasma	40	PC3

## Data Availability

Data will be made available upon reasonable request.

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
