# Peer review of "Machine Learning-Enabled Quantitative Analysis of Optically Obscure Scratches on Nickel-Plated Additively Manufactured (AM) Samples"

_materials, 2023, doi:10.3390/ma16186301_

Round 1
Reviewer 1 Report
Comments of materials-2594760
The main weaknesses of the manuscript:
1. A state-of-the-art review of past works in regard to this study is lacking. Hence, the novelty of the study is not highlighted.
2. Provide a detailed experimental procedure for the study. How many samples were produced? Make a table for this. The chemical constituents of the samples using XRF or EDS or XRD should be supplied and discussed.
3. Indicate the micropores on the SEM of sample 3 and explained in the manuscript.
4. To corroborate a good manuscript, the authors are to add the mechanical properties of the developed samples.
5. Some statements in the conclusion have no result to justify the claim.
Author Response
Reviewer 1
Comment 1: A state-of-the-art review of past works in regard to this study is lacking. Hence, the novelty of the study is not highlighted.
Response: The introduction section has been enhanced with related references. The emphasis and rationale for the paper theme have been highlighted in the revised version.
Comment 2: Provide a detailed experimental procedure for the study. How many samples were produced? Make a table for this. The chemical constituents of the samples using XRF or EDS or XRD should be supplied and discussed.
Response: We appreciate the reviwer's comment for improving the experimental section. Experimental sections have been revised with additional practical details. Please note the major focus of this paper is on the application of ML in analyzing the nickel coatings produced on BJAM. Hence, we focused on elaborating on the postprocessing aspect of scratches made on nickel coating and ML approach used in this paper.
Comment 3: Indicate the micropores on the SEM of sample 3 and explained in the manuscript.
Response: This is a valuable question about the integrity of the films. Electroless nickel coating was typically continuous and solid and did not show noticeable pores. Generally, 20-40 µm nickel films filled undulations on the produced BJAM surface. Due to that reason, we mainly focused on roughness and shape of the scratch. The in-depth SEM-based morphological studies are part of a separate publication that is in progress.
Comment 4: To corroborate a good manuscript, the authors are to add the mechanical properties of the developed samples.
Response: Since we focused on the postprocessing of the fully completed AM part, we did not study mechanical testing like standard Tensile testing. Our external electroless nickel coatings are present on the top and do not impact the bulk mechanical properties; and hence, tensile testing was not attempted. The reviewer has given excellent suggestions about related mechanical testing capable of investigating the postprocessing treatment effect(electroless nickel coating). In the future, Fatigue testing, like mechanical testing methods that are related to the surface or postprocessing, may be conducted. However, Fatigue testing needs extensive work and is beyond the scope of this paper.
Comment 5: Some statements in the conclusion have no result to justify the claim.
Response: We appreciate the reviewer's comment about improving conclusions. We have revised the conclusions by including key results.

Reviewer 2 Report
- The authors only explored one kind of clustering method i.e. k-means clustering. The authors should explore other traditional clustering methods.
- Further, apart from traditional clustering methods, the authors should compare withs state of the art clustering methods such as using neural network clustering. These articles such as https://www.tandfonline.com/doi/abs/10.1080/03772063.2021.1965043?journalCode=tijr20 & https://pubmed.ncbi.nlm.nih.gov/35990143/ elucidate using neural networks for clustering. Further, this kaggle link https://www.kaggle.com/code/loaiabdalslam/clustering-using-deep-learning provides an example of using clustering using deep learning.
- Further, there is explainability lacking in the clustering. The authors should use explainability methods like Shapley (https://www.aidancooper.co.uk/supervised-clustering-shap-values/) to explain the clustering.
- The figures are low resolution especially Figure 2.
Author Response
Please see the attachement.

Reviewer 3 Report
This study aims to investigate the relationship between the relative surface hardness and scratch width of electroplated nickel plating on incrementally produced composite components. To evaluate this relationship, the authors used unsupervised machine learning methods based on K-means clustering. Taguchi Experimental Design (TED) method was applied in the research. In order to reduce surface reflections, a Scanning Electron Microscope (SEM) was used to capture grayscale images and three-dimensional height maps of the scratched Ni-coating samples; this allowed for a rigorous characterization of the scratch widths. The TED approach encompasses the study of various Ni-coating hardness levels for nine different samples and a complex examination of the non-linear effects on this.
However, I recommend that the authors reconsider the following points:
1- The statistical analysis in Figure 4 needs a comprehensive explanation.
2- The resolution of Figure 2 is quite low; therefore, this illustration should be recreated to emphasize the importance of critical points.
3- Table 2 is missing essential information for this study. This table should either be removed or enriched with detailed explanations.
4- The references in the References section should be increased by using up-to-date sources.
5- In the conclusion section of the article, the achievements of the study should be clearly underlined.
Author Response
Reviewer 3
Please note the following review 3 comments are the same as Reviewer 2 and have been answered.
1- The statistical analysis in Figure 4 needs a comprehensive explanation.
2- The resolution of Figure 2 is quite low; therefore, this illustration should be recreated to emphasize the importance of critical points.
3- Table 2 is missing essential information for this study. This table should either be removed or enriched with detailed explanations.
4- The references in the References section should be increased by using up-to-date sources.
5- In the conclusion section of the article, the achievements of the study should be clearly underlined.

Round 2
Reviewer 2 Report
The authors have improved the figure quality and added some explanation for the clustering. The authors have mentioned about their ML skills limitations and plan to add more comparison in their future works.